# Road Traffic Forecast Based on Meteorological Information through Deep Learning Methods

**DOI:** 10.3390/s22124485

**Published:** 2022-06-14

**Authors:** Fernando José Braz, João Ferreira, Francisco Gonçalves, Kawan Weege, João Almeida, Fabiano Baldo, Pedro Gonçalves

**Affiliations:** 1Instituto Federal Catarinense Campus Araquari, Araquari 89245-000, Brazil; fernando.braz@ifc.edu.br; 2Departamento de Electrónica Telecomunicações e Informática e Instituto de Telecomunicações, Universidade de Aveiro, 3810-193 Aveiro, Portugal; joaogferreira@ua.pt (J.F.); fdgoncalves@ua.pt (F.G.); 3Departamento de Ciência da Computação, Universidade do Estado de Santa Catarina, Florianopolis 88035-901, Brazil; kawanweege@gmail.com (K.W.); fabiano.baldo@udesc.br (F.B.); 4Instituto de Telecomunicações, Universidade de Aveiro, 3810-193 Aveiro, Portugal; jmpa@ua.pt; 5Escola Superior de Tecnologia e Gestão de Águeda e Instituto de Telecomunicações, Universidade de Aveiro, 3810-193 Aveiro, Portugal

**Keywords:** weather-based traffic prediction, highway traffic, deep learning, method comparison

## Abstract

Forecasting road flow has strong importance for both allowing authorities to guarantee safety conditions and traffic efficiency, as well as for road users to be able to plan their trips according to space and road occupation. In a summer resort, such as beaches near cities, traffic depends directly on weather conditions, variables that should be of great impact on the quality of forecasts. Will the use of a dataset with information on transit flows enhanced with meteorological information allow the construction of a precise traffic flow forecasting model, allowing predictions to be made in advance of the traffic flow in suitable time? The present work evaluates different machine learning methods, namely long short-term memory, autoregressive LSTM, and a convolutional neural network, and data attributes to predict traffic flows based on radar and meteorological sensor information. The models trained to predict the traffic flow have shown that weather conditions were essential for this forecast, and thus, these variables were employed in the evaluated deep-learning models. The results pointed out that it is possible to forecast the traffic flow at a reasonable error level for one-hour periods, and the CNN model presented the lowest prediction error values and consumed the least time to build its predictions.

## 1. Introduction

### 1.1. Motivation

Intelligent transportation system (ITS) technologies are becoming more pervasive in the context of smart cities and smart roads, not only due to the development of connected and automated vehicles (CAVs) but also through the deployment of connected roadside infrastructures. The latter are equipped with traffic sensors and communication platforms capable of extracting useful information from the road and transmitting it to vehicles in the vicinity and to traffic management centers.

Connected and sensor-equipped roadside infrastructures can assist in both real-time applications, such as joint maneuvers or route planning, traffic analysis and statistics collection for city management or other third parties’ purposes. For instance, a traffic radar on the highway can support a lane merging use case by helping autonomous vehicles join the main lanes. However, it can also count the number of vehicles entering the highway and therefore be used to assist traffic control prediction and congestion control decisions.

The growth of traffic congestion is one of the main challenges faced today by drivers, motorway operators, and city managers. It makes daily travel more complex, with negative impacts on the environment, time, and monetary costs for the users, degrading the traveler experience. This problem can be mitigated by providing traffic flow predictions to the road users, such as the probability of traffic congestion, which can help them to avoid annoying events by rerouting their travels, choosing another means of transport, or changing their trip times. Additionally, traffic flow predictions can be beneficial for road and city operators to implement traffic planning and management strategies.

Traffic flow prediction is a subject addressed by the literature within the traffic state estimation (TSE) problem scope. TSE refers to the process of inference of traffic state variables, namely flow, density, speed, and other equivalent variables, on road segments, using partially observed and noisy traffic data [1]. There is significant research interest for the TSE topic in the ITS area, given its utility for public authorities, road operators and the general public.

In this work, the main motivation lies in the need to estimate the traffic flow on a coastal area in Portugal—a summer resort composed by the beaches of Barra and Costa Nova—that are close to a mid-size city (Aveiro) with the goal of predicting congestion periods, especially during the peak season. This information is extremely helpful for the decision-making process of beach users and city planners. If provided in real-time, it can help alleviate traffic congestion in the area at crowded moments that typically occur during summer, weekends or specific bank holidays. In this scenario, the traffic is directly impacted by weather conditions, variables that should be of great impact on the quality of forecasts. As a result, some research questions can therefore be posed, for instance: Is it possible to create a real-time traffic prediction model that uses weather information? What would be the most suitable prediction model, both in terms of accuracy and in terms of computational performance?

### 1.2. Contribution

The present work evaluated the use of different machine learning (ML) methods and data attributes to predict traffic flows based on traffic radar and meteorological sensor information. For this purpose, real-world datasets from an ITS ecosystem developed in the urban region of Aveiro in Portugal [2] were employed, taking advantage of roadside infrastructure sensors installed in the field.

Therefore, ML methods, namely long short-term memory (LSTM), autoregressive LSTM (AR-LSTM), and a convolutional neural network (CNN), were used to train TSE models to forecast the traffic flow on the two different coastal beaches of Portugal. To construct more precise models, weather conditions are essential to predict traffic flow in the beach regions, because the affluence of the beaches depends on the temperature, wind, solar radiation, and so forth. Therefore, these weather variables are also used as features in the induction of the models.

Approaches that extract dependence between data from historical data by using ML methods and then estimate the traffic state based on real-time data are called data-driven approaches for TSE [1]. Therefore, based on such framing, this work can be considered a case of a data-driven approach for TSE.

### 1.3. Organization

The rest of the paper is organized as follows: Section 2 reviews neural network methods and performance metrics of machine learning models. Then, Section 2.3 details the state-of-the-art on traffic flow and road speed forecasting published in the most recent years. Section 3 presents the proposed method for traffic flow prediction, namely the datasets used for the training, validation, and test of the induced models, including both the road sensor and the meteorological data. This section also explains the methodology of the work by describing the data selection, pre-processing and concatenation processes, and the model training for each of the four different ML methods. After that, Section 4 depicts the outcomes of the experiments, while Section 5 provides a discussion of the obtained results. Finally, Section 6 summarizes the conclusions and future work.

For a better understanding of the manuscript, we defined the following Table 1 with all the acronyms used.

## 2. Background

### 2.1. Neural Networks

In the past decade, deep learning approaches have dominated the bulk of articles/ implementations for the previously mentioned application areas [3], becoming the go-to when dealing with novel forecasting challenges. At its core, deep learning is a subclass of machine learning and is influenced by the structure of the human brain. The primary building blocks of deep learning architectures are neural units. By hierarchically assembling these units (which in turn creates a neural network), in theory, every kind of nonlinear function can be approximated. There are various famous deep learning architectures, such as CNN, graph convolution networks (GCN) and recurrent neural networks (RNN) and their variants, like LSTM. Using these and other similar architectures, it is possible to model spatial dependencies, temporal dependencies or even joint space-time dependencies.

RNNs as well as its variants are neural networks that analyse a sequential input, and they are proficient at modelling the non-linear temporal dependency of traffic data. Because these models rely on the order of data to process data, in turn, one downside is that when modelling extended sequences, their capacity to retain what they learnt before many time steps may deteriorate (vanishing and/or exploding gradient problem). In addition to this, it cannot perform parallel calculations. Awan et al. [4] used LSTM to project traffic flow using time-series traffic flow, atmospheric data, and air pollution acquired from open data sets of Madrid, concluding from their experimental results that the addition of atmospheric and air pollutant information with timestamp improved the overall performance.

CNNs can be used to model both spatial and temporal dependencies. Unlike recurrent models, CNNs generate models for fixed-size contexts, although the actual context size of the network may be readily increased by adding numerous layers on top of one another, which enables exact control over the maximum length of dependencies to be simulated. CNN also permits parallelization for each element in the sequence (optimising GPU usage), since it does not rely on the computation of the preceding time step. Having said this, CNNs also have some disadvantages: they do not encode the position and orientation of objects, they are not able to be spatially invariant to the input data and they require a large training data to be accurate. An example of the usage of this method is present in [5], where Gehring et al. propose the very first fully convolutional model for sequence-to-sequence learning that surpasses robust recurrent models on extremely large benchmark data sets by an order of magnitude in time complexity.

### 2.2. Performance Metrics

Before presenting the most commonly used metrics to assess prediction models, it is important to distinguish the two different types of predictions performed by ML methods. They are regression and classification. Roughly, it can be said that regression is related to the prediction of continuous values, in contrast to classification, which is devoted to predicting discrete labels. Therefore, as the prediction of traffic flow is concerning the forecasting values that vary in a continuous matter, this work addresses a regression problem.

To test the performance of the regression model proposed in [6], Hou et al. used five performance metrics: mean absolute percentage error (MAPE), symmetric mean absolute percentage error (SMAPE), mean absolute error (MAE), mean square error and root-mean-square error (RMSE). The authors considered that these metrics reflect the difference between actual and predicted values, fulfilling their purpose. In [7], Deb et al. only used RMSE, while Yin et al. [8] used MAE, RMSE and MAPE. Shahriar et al. [9] used the metrics of RMSE, MAE, SMAPE and the coefficient of determination ($R^{2}$) to evaluate the performance of the regression models. Generally, small values of RMSE, MAE, and SMAPE demonstrate accurate predictions. The coefficient of determination is a measure of the goodness of fit of the data and is usually between zero and one. If the value of the coefficient of determination is equal to one, we have a perfect prediction. Generally, a higher value of the coefficient of determination demonstrates a better performance [9]. Badii et al. [10] only used the performance metric mean absolute scaled error (MASE). However, according to these authors, the lack of data can make it difficult to produce results, affecting the value of MASE. According to the authors, the data collected by sensors are the most sensitive to failures in data collection, admitting that this failure may result from the sensors themselves or from connection problems. To combat this problem, a Kalman filter was used, a method that produces estimations of unknown data, given the measurements observed in the temporal space.

Given the abundance of traffic forecasting methods, performance metrics for each method also emerge. Although RMSE and MAPE are metrics widely used to evaluate the performance of a model, they do not provide comparable measures when the models are divergent, for example, a neural network and an autoregressive integrated moving average (ARIMA), or when the input datasets of the models are completely different [11]. In the case of network-wide models, errors can propagate through time and space of the network, and therefore, a space-time correlation between successive predictions can help to assess performance. Despite contemporary studies using RMSE and MAPE to evaluate performance, the necessity of evaluating datasets, environments and performance metrics remains present, as identified by Lana et al. [11].

Novakovic et al. studied the problem of evaluating the different classification models that are used in ML [12]. It is necessary to evaluate these models in order to find the optimal solution for the classification models generated in the construction process. According to Novakovic et al., there are different measures to evaluate the performance of a model, and the most used criterion is the calculation of accuracy.

Classification is one of the most common tasks in ML. It is based on the search for similarities between objects, and the similarities between both are determined by analyzing their characteristics. In a classification problem, the number of classes is previously known and limited. In the evaluation of classification methods, if it predicts a class that is different from the current class, then we are facing a classification failure. This notion of failure results in the accuracy formula, which can be defined as the number of correctly classified cases out of the total number of cases. However, the accuracy does not reflect the differences between error types, which is a disadvantage in using this formula. Furthermore, the accuracy is dependent on the class distribution in the dataset [12].

It is necessary to distinguish different types of error because the consequences resulting from this error may be different. For example, in medicine, if a system is in charge of classifying a cancer case as positive or negative, the result of this classification leads to two types of consequences. In case the system identifies a positive patient as negative, the error is more important than in the reverse situation, since doctors will not consider the patient to be sick and, consequently, will not apply any treatment [12].

### 2.3. Related Work

A lot of work has been done on traffic and road speed forecasting in recent years. Yi et al. [13] developed and automated a framework for tunning traffic dataset hyperparameters in order to reduce time-consuming learning tasks. The proposal describes a framework named HyperNet that uses Bayesian optimization and meta-learning for the automated hyperparameter search process. Furthermore, they implement a deep learning model with a long short-term memory network based on the HyperNet framework and present a learning temporal variation of traffic datasets at main regions of highway traffic systems. Their platform evaluation is performed with data they collected from the Korean highway system, but it does not include any weather information.

Sadeghi-Niaraki et al. presented a short-term traffic flow prediction model [14] based on the modified Elman recurrent neural network (ERNM) model to improve traffic prediction model precision. They used a modified ERNM method optimized through a genetic method, and they considered weather conditions, weekday, hour and day’s classification to forecast the vehicle velocity in Tehran streets and highways, but they did not forecast traffic fluxes. The traffic dataset was collected from online Google Map API service for 139 routes in 7 Tehran districts. The reported results confirmed the superior performance of the proposed traffic condition prediction model over several alternative methods.

Hu et al. proposed a traffic speed prediction model [15] based on a temporal convolution network (TCN) and a graph convolution network (GCN). Their approach uses the TCN to complete the extraction of time dimension and local spatial dimension features and the GCN to extract the topological relationship between road nodes in order to attain global spatial dimension feature extraction. They combine spatial and temporal features with road parameters to improve short-term traffic speed predictions. The reported experimental results used an open dataset with 157 sensor stations from the United States highway dataset PeMS [16] and showed that their model obtained the best performance under various road conditions compared with eight baseline methods. Moreover, results showed that their prediction error was reduced by at least 8%, keeping high effectiveness and stability. The study dis-considers traffic volume, and it could not be applied to our reduced highway topology.

Wei et al. proposed a traffic flow prediction method [17] called autoencoder long short-term memory (AE-LSTM) to improve the accuracy of traffic flow prediction. The method presented was used to obtain the internal relationship of traffic flow by extracting the characteristics of upstream and downstream traffic flow data. The LSTM network utilizes the acquired characteristic and historical data to predict complex linear traffic flow data. The experimental results obtained using an unspecified PeMS dataset [16] showed that the AE-LSTM method had higher prediction accuracy, since the mean relative error (MRE) of the AE-LSTM was reduced by 0.01 compared with the previous prediction methods. Moreover, the AE-LSTM method also offered good stability as it showed a prediction error and fluctuation of the AE-LSTM method that was small for different stations and different dates. Reports show an average MRE for AE-LSTM prediction results of 0.06 for six different days.

Simunek et al. proposed a long-term traffic speed prediction ensemble model [18] using country-scale historic traffic data from Czech Republic roads. Their model combined parametric and nonparametric approaches in order to acquire a good-quality prediction that can enrich available real-time traffic information. The model was conceived to be used in the improvement of navigation through waypoints (e.g., delivery services, goods distribution, police patrol) and in the estimation of arrival time. Model validation was carried out using 66% of all roads in the Czech Republic and predicted traffic speed in the period of 1 week. The validation tests showed that the average speed prediction at a given hour could make predictions with a mean absolute error of 4.67 km/h, which looks promising in terms of long-term speed information prediction using large-scale spatial and temporal data.

Sun et al. proposed an XGBoost-based approach to predict highway traffic flow [19]. Their method started by dividing highway segments and using cameras to directly cover road sections. Then, a section-flow calculation method was used to predict the traffic state. Moreover, they improved that information with toll station entrance and exit information and with plate number recognition information. Then, they applied an improved XGBoost-based spatio-temporal method with the early stopping rounds adjustment mechanism (EAM) optimization mode to predict the traffic flow of the segmented highway, which considers multiple-step short-term and long-term prediction, the influence of nonrecurrent incidents, and the spatial interaction of sophisticated staggered sections. The dataset was created from 1 September to 19 November 2019 in a Shaoxing, China road network and it consists of 23,040 intervals over 80 days in total. The paper additionally compared traffic forecasts for 5, 10, 15, 20, 25 and 30 min intervals with alternative traffic forecasting methods such as CNN, CNN-lag, LSTM, LSTM-lag, random forest (RF), RF-lag, seasonal autoregressive integrated moving average (SARIMA), SARIMA-lag, XGBoost-I, XGBoost-I-lag, XGBoost-S, and XGBoost-S-lag. XGBoost-I-lag achieves the best performance compared with XGBoost-S series models and other baseline models. The test results confirmed that the missing data greatly affects the traffic flow prediction results in the XGBoost-I-lag, and that, except for SARIMA, the spatial lag input of all methods is better than the ordinary input. The authors also observed that the identified spatio-temporal lag strategy is extremely necessary in highway traffic prediction.

Guo et al. proposed an optimized graph convolution recurrent neural network for traffic prediction with the aim of better exploiting the representation of spatial and temporal information [20]. They learned spatial-temporal features of the traffic data by a graph convolution gated recurrent unit. In their experiments, they used several datasets: a travel time/speed dataset that covers the northwestern part of the Washington, DC, USA metropolitan area, collected in the summer of 2016; a travel time/speed dataset that covers the center city of Philadelphia, collected in the summer of 2016; and a traffic flow dataset in PeMSD4 that covers the San Francisco Bay Area, collected from January to February in 2018. They reported that their method was validated and proven effective and accurately predicted traffic data in the future 15 and 30 min.

Chiabaut et al. presented a method for real-time traffic and travel time estimation for highways [21]. The proposal included a previous analysis consisting of a historical dataset observation day using two methods: a Gaussian mixture model and a k-means method. The produced clustering results revealed that congestion maps of days of the same group have substantial similarity in their traffic conditions and dynamic. Then, consensus days were identified in each cluster as the most representative day of the community according to the congestion maps. Finally, the traffic congestion propagation and travel times prediction was carried out based on the historical data information. The dataset used was created in a M6 highway used to access Lyon’s city center through a tunnel, created by 9 detectors of a 7 km long section. The work reported 40% to 79% arrival prediction precision below 2 min, and 68% to 87% below the 3 min window.

Zhang et al. proposed a graph convolutional method based on deep learning for highway traffic toll flow prediction [22]. They considered spatio-temporal and external factors such as weather conditions and date type and used a GCN to extract spatial features from their model. In their work, they used a dataset created using traffic flow of all 269 highway toll stations in Henan Province, China and compared their accuracy with other models. They explained their higher prediction accuracy as due to the use of a GCN to obtain the spatial factors of the highway network and reported that their model behaves better than other models such as K-nearest neighbors (KNN) and LSTM in the metrics RMSE, MAE, and MAPE.

Table 2 summarizes the information about the related work documented in terms of the dataset used, the processing techniques that were used and the learning process objective. This analysis allows to perceive different forecasting objective [13,15,21,22], works that did not sufficiently detail the dataset used or the conditions of use [17,19] or were developed for very different road models [14,18], so our option was to test several deep learning methods in order to evaluate them in terms of accuracy and efficiency.

## 3. Proposed Traffic Flow Prediction Method

In order to tailor the method to predict traffic flow for the coastal beaches of Portugal, firstly, we performed the activity of selecting the available data in the surrounding area of the two selected beaches. Then, we assessed whether the available data were relevant enough to handle the prediction models’ induction.

The selected datasets are composed of two different sources: radars and meteorological data. Figure 1a,b present the location of the radars and meteorological stations, respectively.

Figure 2 presents the complete method designed to handle the induction of the models for traffic flow prediction on Portugal beaches. It starts with the data selection (Figure 2a) applied to telemetry and meteorological data.

The Telemetry dataset contains records produced by parking sensors and radars, and these records are separated into two different datasets: Parking and Radars. After that, the data preprocessing step starts (Figure 2b) to prepare the meteorological and radars data before the mining process.

In the sequence, the preprocessed datasets are concatenated into a single one (Figure 2c), which will be the data source for the mining tasks.

Finally, during the model training phase (Figure 2d), the data were processed by the data mining techniques, producing the models throughout LSTM and CNN.

The following subsections describe in detail each one of the steps presented in the methodology.

### 3.1. Data Selecting

The original Telemetry dataset contains more than 170 million records (170,158,409) considering the years of 2019, 2020, and 2021 and is composed of parking sensors and radars data. For the goal of this work, just radar data were selected, resulting in 155,432,185 records.

Table 3 presents the attributes of the original data. Each record is produced at a frequency of 100 ms and contains the identification of the moving object, ID and coordinates of the radar, timestamp, and the *x*-*y* axis speed component.

The meteorological dataset used in this work was provided by the Portuguese Institute of Sea and Atmosphere (IPMA) [23]. IPMA maintains an up-to-date climate dataset with information such as air temperature, wind speed, and direction, light radiation and so forth. Figure 1b presents the two meteorological stations near to the radars used in this work. One is placed at the University in Aveiro, and the other is at Dunas de Mira.

Table 4 presents the attributes of the meteorological data. Each record is produced at a frequency of ten minutes.

### 3.2. Data Preprocessing

The radar data are produced at a frequency of 100 ms; however, the meteorological data are generated at a frequency of 10 min. The adjustment of the difference in the granularity (100 ms × 10 min) among the two datasets is one of the tasks executed in the preprocessing phase.

Before adjusting the granularity, other derived radas data were produced; *year, month, day, hour, weekday*, and *minute* attributes were calculated from the *timestamp.* Addutionally, the *xSpeed* and *ySpeed* attributes result in the *Speed* measure. Negative values for *Speed* represent the measure of the speed of an object approximating the radar, and positive values represent the object moving away; the *in_out* logical attribute stores this situation.

Using the identification, speed, and direction of the moving object, for each radar, it is possible to compute the quantity and speed (*maximal*, *mean*, and *minimal*) at the level of the *radar*, *year*, *month*, *day*, *hour* and *minutes* (ten minute intervals). Table 5 presents the resulting format of the processing radar data, and each record represents measures aggregated for ten minutes at the hour.

Figure 1a presents the localization of the radars named “*pasmoradar03pontepraias*”, “*pasmoradar02poste12*” and “*pasmoradar01riativa*”. The first one occurs before entering the bridge; the second one is in the interconnection segment between Barra and Costa Nova, and the third one is at the urban limit to the south of Costa Nova.

This work aims to develop a model to forecast the traffic flow in Barra and Costa Nova, considering the meteorological environment and level of speed vehicles approximating and leaving the radars. Then, two new measures were computed to represent the traffic flow in the regions *TF_Barra* and *TF_Costa*.
(1)$$TF\_Barra=\left(QAR_{1}+QDR_{2}\right)-\left(QDR_{1}+QAR_{2}\right)$$
(2)$$TF\_Costa=\left(QAR_{2}+QAR_{3}\right)-\left(QAR_{2}+QDR_{3}\right)$$
where $QAR_{i}$ = the quantity of objects approximating the radar *i*, and $QDR_{i}$ = the quantity of objects detaching from the radar *i*, computed by
(3)$$QAR_{i}=\sum_{i=1}^{3}\sum_{j=0}^{50}obj\_count\_A_{i,j}$$
(4)$$QDR_{i}=\sum_{i=1}^{3}\sum_{j=0}^{50}obj\_count\_D_{i,j}$$
with *i* as the identification of the radar, *j* as the interval minute (0, 10, …, 50), *obj_count_A* as the value of the *count_obj* attribute where *in_out* = 1, and *obj_count_D* as value of the *count_obj* attribute where *in_out* = 0.

Therefore, *TF_Barra* and *TF_Costa* can be positive or negative values. A positive *TF* value represents an increase in the traffic flow for that region, and negative values can express a reduction. Additionally, new speed measures were computed considering the movement of approximation and detaching for each radar.
(5)$$Speed\_med\_AR_{i,j}=\frac{1}{n}\sum Speed\_med_{i,j}$$
with *n* as the quantity of records representing the Speed_med attribute with in_out = 1.
(6)$$Speed\_med\_DR_{i,j}=\frac{1}{n}\sum Speed\_med_{i,j}$$
with *i* as the identification of the radar, *j* as the interval minute, and *n* as the quantity of records representing the Speed_med attribute with in_out = 0.
(7)$$\begin{array}{c} Speed\_max\_AR_{i,j}=\max\left(Speed\_max_{i,j}\right) \end{array}$$
(8)$$\begin{array}{c} Speed\_max\_DR_{i,j}=\max\left(Speed\_max_{i,j}\right) \end{array}$$
(9)$$\begin{array}{c} Speed\_min\_AR_{i,j}=\min\left(Speed\_min_{i,j}\right) \end{array}$$
(10)$$\begin{array}{c} Speed\_min\_DR_{i,j}=\max\left(Speed\_min_{i,j}\right) \end{array}$$

In Equations (Equation 5)–(Equation 10), *i* represents the identification of the radar, and *j* represents the interval minute. Equations (Equation 7) and (Equation 9) consider the records with in_out = 1, while for Equations (Equation 8) and (Equation 10), in_out = 0. Table 6 presents the final processed radar data.

Figure 1b presents the two meteorological stations collecting environment measures. Each record represents one observation every ten minutes for each hour of the day. The processing of the meteorological data was focused on two goals: (1) correcting the lack of measures of the stations, and (2) transforming the wind direction values from degree to cardinal values.

In situation (1), some records are incomplete in one station but complete in another. In this case, considering that the meteorological stations are close and the variation of the measures is not significant, the procedure was to complete the fault value by using the value of the other station.

Wind direction values recorded in degree are a potential problem at the moment to execute the data mining tasks (situation 2). For instance, the direction North (cardinal) represents the degree interval between 348.75 and 11.25. Therefore, considering the direction, a value of 5 degrees represents the same direction as a value of 359 degrees. However, the two values could represent two different environments in the execution of mining tasks. The solution was to transform the values into eight cardinal points (*N, NE, E, SE, S, SW, W, NW*) defined by a 45 degree interval.

Table 7 presents the processed meteorological data. The next step is to concatenate the two datasets: radars and meteorological.

### 3.3. Concatenating

According to Figure 2, the final step of preparing data is to combine the two datasets to produce a single set to be mined. The concatenation was executed considering the following attributes as indexes: ***year***, ***month***, ***day***, ***hour***, and ***minute*** (intervals). Therefore, each record of the final dataset is composed of index columns and the attributes representing the radar and meteorological measures. The final dataset has 74,305 records ordered and summarized by index columns and is considered a multivariate timeseries. Table 8 presents the final dataset.

### 3.4. Model Training

The dataset was not randomly shuffled before splitting. The final dataset that was preprocessed in the previous steps was divided into training, validation, and testing sets, composed of 70, 20, and 10% of time-ordered records. The time ordering is necessary to build windows of consecutive records, and it is a way to execute training, evaluation, and testing steps more realistically due to the time series form of the data. In addition to the radar information, the dataset contains contextual data [3]: weekday, temperature, solar radiation, speed and wind direction, and precipitation. These attributes can improve the performance of the predictive models; however, they are related to the year’s seasons. Therefore, to ensure that the model considers records of all seasons, the training set was defined with 70% of the dataset; it is equivalent to the interval between January and December of 2020.

Figure 3 presents the complete dataset divided into subsets; the *x*-axis represents the time and the *y*-axis the traffic flow values. Figure 4 presents similar data distributions of the subsets.

This work aims to develop regression models for forecasting the traffic flow in the Barra and Costa Nova regions. Regression models consider predictor and target variables; in this case, *TF_Barra* and *TF_Costa* dataset attributes were the target variables, and others, e.g., *month, day, weekday, hour, minute, temperature, speed*, and *wind*, were used as predictors. Each record presents the values (predicted and targeted) aggregated by ten-minute intervals; therefore, a sequence of six records represents the behavior of that environment during one hour.

We conducted experiments using three deep-learning regression methods (*LSTM, AR-LSTM, CNN*) to forecast the traffic flow; *mean absolute error* (*MAE*) was used as a performance metric. Each model was constructed considering the best configuration, defined as the result of an optimization phase by using the hyperparameter options (*neurons, activation function, optimizer function, dropout, batch size, filter map, and kernel size)*. Each model was trained to find the best number of epochs, considering the limit of 200 epochs. For each method, six different time intervals were considered: (1,1), (2,2), (3,3), (4,4), (5,5), and (6,6), where (1,1) means one previous record (ten minutes) and one (ten minute) traffic flow predicted. Therefore, this work presents results to forecast the traffic flow between ten and sixty minutes (one hour) considering a multi-step model. In a multi-step model, the proposal is to build a model where it is possible to consider the changing of the input features along the time to forecast the sequence values. Figure 5 presents the single and multi-step models.

The CNN model was defined with a convolutional layer with 64 filter maps, a Relu activation function, and a kernel size of the same length as the time interval value, e.g., kernel size = 6, where the time interval was 6 (one hour). A dense layer was used with the number of nodes defined by the product between the time interval value and the number of features, e.g., considering the dataset with 43 features and one hour as the period to forecast, the number of nodes was 258 (43 × 6). Finally, the last layer converted the format to present the predicted traffic flow values.

The LSTM method has been used to analyze time-series datasets; the proposal is to accumulate the internal state during the time interval and then compute the forecast for the next time interval. In this work, the LSTM model had a first layer with 32 neurons and was configured to return the output at the final time step. There was a dense layer with the same configuration as the CNN in the sequence. The number of nodes was defined by the product between the time interval value and the number of features, e.g., considering the dataset with 43 features and one hour as the period to forecast, the number of nodes was 258 (43 × 6). Finally, the last layer converts the format to present the predicted traffic flow values.

Finally, the AR-LSTM model decomposed the prediction into individual time steps. The approach was to use each model output to feed back into itself. Therefore, the forecasting could be done considering the previous result. In the same way, as in the LSTM model, the LSTM layer contains 32 neurons.

## 4. Results

Tests were carried out for the data relating to Barra and Costa Nova beaches on an Intel(R) Xeon(R) CPU E5-2620 v4, with 8 cores, with a CPU frequency equal to 2100 MHz and RAM memory equal to 16 GB running an Ubuntu 20.04.4 (LTS). The tests were carried out as remotely scheduled via ssh in a server distribution of the operating system. In addition, equivalent intervals were used (input size equal to output size), starting with one ten-minute block to six ten-minute blocks (sixty minutes); that is, for a ten-minute break, we prediced the next ten minutes, while for a twenty-minute break, we predicted the next twenty minutes, until we reached sixty minutes. The methods used were CNN, LSTM and AR-LSTM. For each interval, the tests were repeated ten times, evaluating the MAE and the execution time. Subsequently, the mean and standard deviation of each of these metrics were calculated.

Analyzing the results of the tests performed with the CNN method on Barra and Costa Nova beaches, presented in Table 9, we can conclude that:Barra beach: the test with the best result for the MAE average corresponds to the test with input and output equal to ten minutes; the test with the best result in the average execution time corresponds to the interval with input and output equal to ten minutes;Costa Nova beach: the test with the best result for the MAE average corresponds to the test with input and output equal to ten minutes; the test with the best result in the average execution time corresponds to the interval with input and output equal to sixty minutes.

Since the tests for input and output equivalent to ten minutes are the ones with the best performance (smallest MAE and shortest execution time) in the case of Barra beach and the ones with the lowest MAE in the case of Costa Nova beach, we could consider that this would be the best option. By analyzing the Table 9, it is possible to conclude that the error and the execution time increase as the size of the input and output increases.

By analyzing the results of the tests performed with the LSTM method to Barra and Costa Nova beaches, presented in Table 10, we can conclude that:Barra beach: the test with the best result for the MAE average corresponds to the test with input and output equal to thirty minutes; the test with the best result in the average execution time corresponds to the interval with input and output equal to ten minutes. However, despite the best MAE average being present in the tests for equal input and output at thirty minutes, tests with equivalent input and output equal to sixty minutes have a smaller standard deviation;Costa Nova beach: the test with the best result for the MAE average corresponds to the test with input and output equal to ten minutes; the test with the best result in the average execution time corresponds to the interval with input and output equal to ten minutes.

Since the tests for input and output equivalent to ten minutes are the ones with the best performance (smallest MAE and shortest execution time) in the case of Costa Nova beach and the fastest in the case of Barra beach, we could consider that this would be the best option. Ignoring the execution time, we could consider other options, for example, the option where the input and output are equal to thirty minutes. Analyzing the data from Barra beach, it is possible to observe that this option has a lower MAE. Finally, it is possible to state that the MAE is between 13% and 15% and that the execution time is between 15 and 32 min, which encourages us to think that as the input and output intervals increase, the execution time increases, and the MAE remains within the usual values (13–15%).

Analyzing the results of the tests performed with the AR-LSTM method on Barra and Costa Nova beaches, presented in Table 11, we can conclude that:Barra beach: the tests with the best results for the MAE average are very similar to the results obtained for the LSTM method, with the tests with input and output equal to thirty and forty minutes containing the lowest MAE (with the forty minutes tests having the lowest overall); the test with the best result in the average execution time corresponds to the interval with input and output equal to twenty minutes. Similarly, the tests with input and output equal to forty minutes have the smallest standard deviation;Costa Nova beach: the test with the best result for the MAE average corresponds to the test with input and output equal to thirty minutes; the test with the best result in the average execution time corresponds to the interval with input and output equal to twenty minutes. It is notable that for both the Barra and Costa Nova beaches, the intervals with the best results, for both MAE average and average execution time, are the same.

## 5. Discussion

TSEs are a very important forecasting tool insofar as they allow local authorities to measure the demand and preparation of public transport and population support infrastructure, and on the other hand to adapt people’s behavior, such as changing routes, the use of public transport or even changing travel times according to forecasts. In the present work, a TSE was developed to predict the access traffic to the beaches of Barra and Costa Nova based on meteorology that uses the datasets created within the scope of the PASMO project and by the IPMA. In the learning process, several methods were tested in order to choose the best results, both in terms of error and in terms of complexity. Different learning and forecasting periods were also used in order to optimize the periods that maximized the overall learning results.

CNN tests for Barra traffic allowed us to verify that the MAE average corresponds to the test with input and output equal to ten minutes and the test with the best average execution time of intervals with input and output equal to ten minutes as well. Costa Nova traffic tests confirmed 10 min intervals allow the best MAE results, but obtained a better execution times for sixty minutes intervals.

LSTM method tests for Barra traffic allowed us to obtain the lowest MAE for thirty minutes intervals; however, tests with equivalent input and output equal to sixty minutes have a smaller standard deviation. The lowest average execution time was obtained for ten minute intervals. Costa Nova tests showed the best results for MAE and execution time for 10 minut intervals. LSTM tests allowed us to obtain MAE values between 13% and 15% under execution times between 15 and 32 min, which leaves us to think the interval increase keeps MAE values within the same interval (13–15%) despite increasing the execution time, and thus, to consider the increase as useless.

Tests with AR-LSTM for Barra traffic allowed us to obtain MAE values similar to those obtained for the LSTM method for thirty and forty minute intervals and lower average execution times for twenty minute intervals. Costa Nova tests had the best MAE for thirty minute intervals and better execution times for twenty minute intervals.

Figure 6b plots the test results for Barra traffic. By analyzing it, we can conclude that, for Barra beach, if the input and output intervals increase, the MAE of the LSTM method remains constant, the MAE of the AR-LSTM method decreases between twenty and forty minutes (increases between forty minutes and sixty minutes) and the MAE of the CNN method increases. Furthermore, it is also possible to conclude that if the input and output intervals increase, the execution time of the LSTM and AR-LSTM methods increases, and the execution time of the CNN method remains constant.

These results are equivalent to the results of Costa Nova beach, as illustrated in Figure 7b.

A global analysis of the results presented in the graphs of Figure 6b and Figure 7b allow us to foresee the advantage of using the CNN method for the implementation of a traffic forecasting mechanism under these conditions.

Additionally, and considering the values of the methods’ execution times, the results allow us to foresee promising results regarding the creation of a real-time traffic forecasting system.

## 6. Conclusions

In the present study, a TSE was developed for the prediction of motorway traffic based on weather data, and several prediction methods were tested in terms of MAE and execution time. The method with the best performance was CNN, both in terms of error and in terms of execution time, having obtained the best results for learning and forecasting intervals of 10 min. The results seem very favorable both in terms of forecast error and in terms of calculation complexity and allow us to envisage its use by the community through real-time information.

As next steps in terms of work, a real-time TSE should be built in order to allow the continuous updating of the learning model with traffic and meteorology data and to provide an updated traffic forecast to the project dashboard. In terms of the performance evaluation of machine learning methods, we plan to test the performance of current methods used with larger learning/forecasting intervals and compare the results with XGBoost approaches.

## Figures and Tables

**Figure 1 sensors-22-04485-f001:**
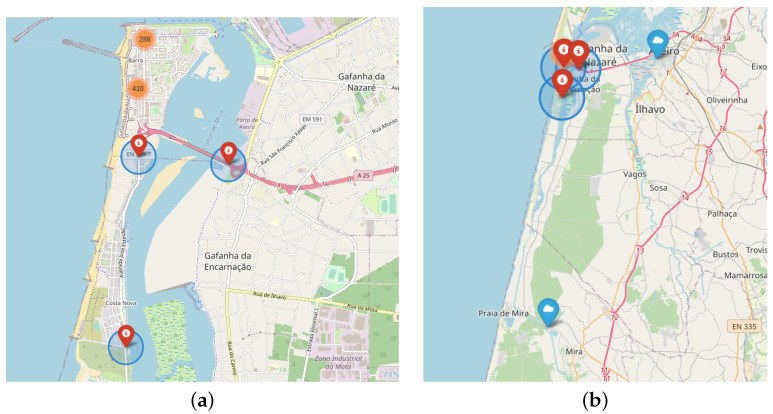
PASMO radar and meteorological station locations: (**a**) Radars; (**b**) Radars and meteorological stations.

**Figure 2 sensors-22-04485-f002:**
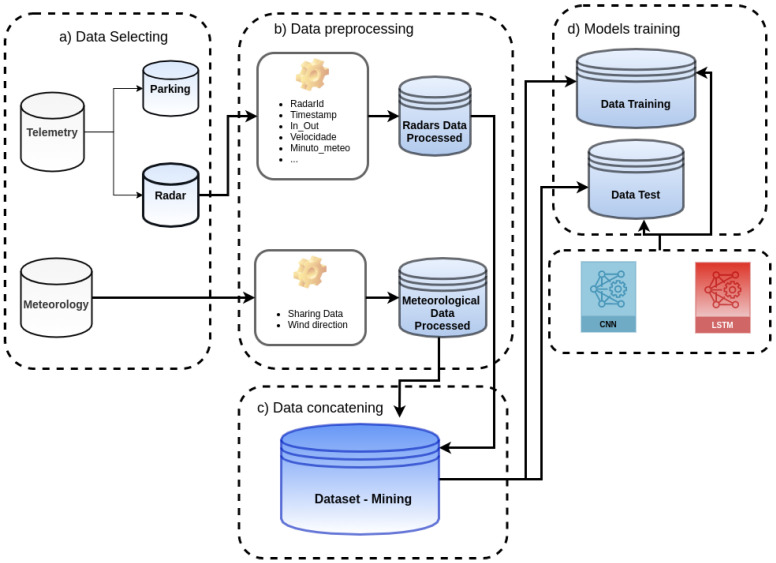
Methodology for data preparation and model training.

**Figure 3 sensors-22-04485-f003:**
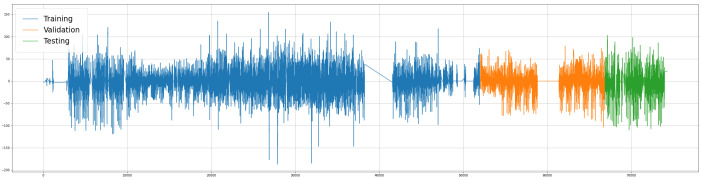
Time-ordered subsets.

**Figure 4 sensors-22-04485-f004:**

Traffic flow: (**left**) training; (**middle**) validation; (**right**) testing.

**Figure 5 sensors-22-04485-f005:**
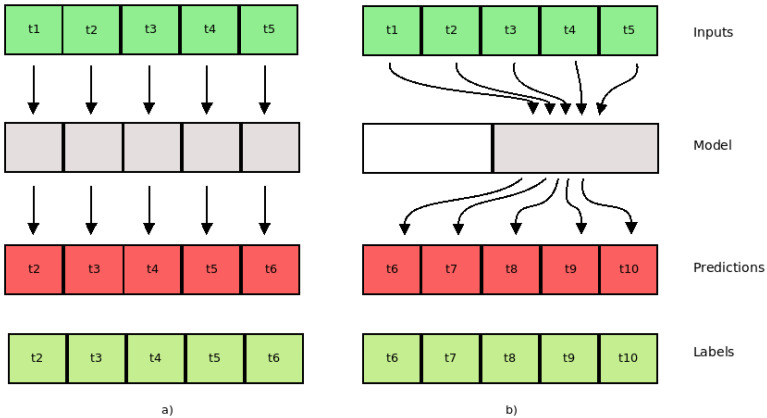
(**a**) Single step; (**b**) multi step.

**Figure 6 sensors-22-04485-f006:**
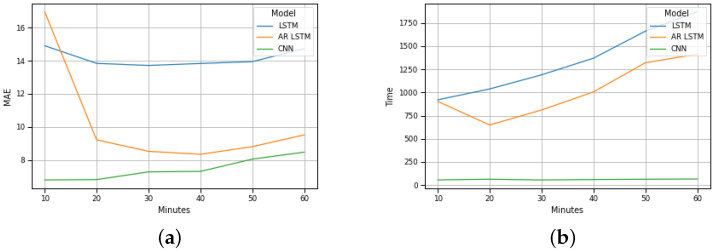
Barra prevision: (**a**) MAE evolution; (**b**) Execution time evolution.

**Figure 7 sensors-22-04485-f007:**
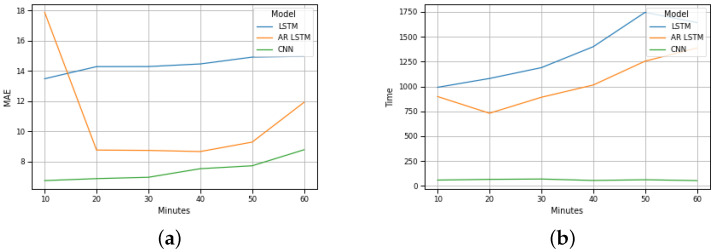
Costa Nova prevision: (**a**) MAE evolution; (**b**) Execution time evolution.

**Table 1 sensors-22-04485-t001:** Abbreviation table.

Acronym	Full Form
*AE-LSTM*	Autoencoder long short-term memory
*ARIMA*	Autoregressive integrated moving average
*AR-LSTM*	Autoregressive long short-term memory
*CAVs*	Connected and automated vehicles
*CNN*	Convolutional neural network
*EAM*	Early stopping rounds adjustment mechanism
*ERNM*	Elman recurrent neural network
*GCN*	Graph convolution network
*ITS*	Intelligent transportation systems
*KNN*	K-nearest neighbor
*LSTM*	Long short-term memory
*MAE*	Mean absolute error
*MAPE*	Mean absolute percentage error
*MASE*	Mean absolute scaled error
*ML*	Machine learning
*MRE*	Mean relative error
*RF*	Random forest
*RMSE*	Root-mean-square error
*RNN*	Recurrent neural network
*SMAPE*	Symmetric mean absolute percentage error
*TCN*	Temporal convolution network
*TSE*	Traffic state estimation

**Table 2 sensors-22-04485-t002:** Related work summary.

Ref.	Data Source(s)	Processing Techniques	Application	Year	Limitations
[13]	Datasets from Korean highway system	Bayesian optimization and meta-learning	Hyperparameter tunning for traffic prediction	2020	Different purpose
[14]	139 routes in 7 districts in Tehran	Modified Elman recurrent neural network model	Traffic forecast	2020	Very different route topology
[15]	157 sensor stations from USA highway dataset	Temporal convolution network (TCN) and a graph convolution network	Traffic speed prediction	2021	Different purpose
[17]	PeMS unspecified dataset	AutoEncoder and LSTM	Traffic flow prediction	2019	No information about dataset
[18]	37,002 km of roads Czech Republic.	Composed model: case-based model, linear regression and fallback	Traffic flow prediction	2020	Very different route topology
[19]	Shaoxing, Zhejiang Province, China road network traffic from 1 September to 19 November 2019	XGBoost-based spatio-temporal method with the EAM	Traffic flow	2021	No information about dataset
[20]	Travel time/speed dataset from northwestern part of the D.C. travel time/speed dataset, from Philadelphia center traffic flow dataset in PeMSD4, and from San Francisco Bay Area	Graph convolution recurrent neural network spatial-temporal features of the traffic data by a graph convolution gated recurrent unit	Traffic prediction	2020	Very different dataset
[21]	9 detectors of a 7 km long section in M6 highway accessed by Lyon’s city center tunnel	Gaussian mixture model and a k-means method	Real-time traffic and travel time estimation	2021	Different aims
[22]	269 highway toll stations in Henan Province	Convolutional method based on deep learning	Highway traffic toll flow prediction	2021	Different aims and dataset

**Table 3 sensors-22-04485-t003:** Original data attributes.

Attribute	Content
*id*	Object ID
*timestamp*	Record timestamp
*radar_id*	Identification of the radar
*radar_lat*	Latitude radar coordinate
*radar_lon*	Longitude radar coordinate
*xSpeed*	*X*-axis speed component
*ySpeed*	*Y*-axis speed component

**Table 4 sensors-22-04485-t004:** Meteorological data.

Attribute	Contents
*Estação*	Station ID
*ano*	Year
*mês*	Month
*dia*	Day
*hora*	Hour
*minuto*	Minute
*t_med*	Mean temperature
*t_max*	Maximum temperature
*t_min*	Minimum temperature
*dd_med*	Mean wind direction
*dd_ffx*	Maximum wind direction
*ff_med*	Mean wind speed
*ff_max*	Maximum wind speed
*pr_qtd*	Rainfall
*rg_tot*	Solar radiation

**Table 5 sensors-22-04485-t005:** Processed radar data.

Attribute	Contents
*timestamp*	Record timestamp
*ano*	Year
*mês*	Month
*dia*	Day
*hora*	Hour
*minuto*	Minute (ten minute intervals)
*radar_id*	Identification of the radar
*Speed_med*	Speed mean
*Speed_max*	Speed maximal
*Speed_min*	Speed minimal
*obj_count*	Quantity of objects
*in_out*	Movement

**Table 6 sensors-22-04485-t006:** Final Processed Radar Data.

Attribute	Contents
*ano*	Year
*mês*	Month
*dia*	Day
*hora*	Hour
*minuto*	Minute (ten-minute intervals)
*Speed_med_$AR_{i}$*	Speed mean approximating radar *i*
*Speed_max_$AR_{i}$*	Speed maximal approximating radar *i*
*Speed_min_$AR_{i}$*	Speed minimal approximating radar *i*
*Speed_med_$DR_{i}$*	Speed mean detaching radar *i*
*Speed_max_$DR_{i}$*	Speed maximal detaching radar *i*
*Speed_min_$DR_{i}$*	Speed minimal detaching radar *i*
*TF_Barra*	Traffic flow at Barra
*TF_Costa*	Traffic flow at Costa

**Table 7 sensors-22-04485-t007:** Processed Meteorological data.

Attribute	Contents
*ano*	Year
*mês*	Month
*dia*	Day
*hora*	Hour
*minuto*	Minute
*t_med*	Mean temperature
*t_max*	Maximum temperature
*t_min*	Minimum temperature
*ff_max*	Maximum wind speed
*pr_qtd*	Rainfall
*rg_tot*	Solar radiation
*ff_med*	Mean wind speed
*dd_card*	Wind direction—cardinal points

**Table 8 sensors-22-04485-t008:** Final Dataset.

Attribute	Contents
* **ano** *	Year
* **mês** *	Month
* **dia** *	Day
* **hora** *	Hour
* **minuto** *	Minute (ten-minute intervals)
*t_med*	Mean temperature
*t_max*	Maximum temperature
*t_min*	Minimum temperature
*ff_max*	Maximum wind speed
*pr_qtd*	Rainfall
*rg_tot*	Solar radiation
*ff_med*	Mean wind speed
*dd_card*	Wind direction—cardinal points
*Speed_med_$AR_{i}$*	Speed mean approximating radar *i*
*Speed_max_$AR_{i}$*	Speed maximal approximating radar *i*
*Speed_min_$AR_{i}$*	Speed minimal approximating radar *i*
*Speed_med_$DR_{i}$*	Speed mean detaching radar *i*
*Speed_max_$DR_{i}$*	Speed maximal detaching radar *i*
*Speed_min_$DR_{i}$*	Speed minimal detaching radar *i*
*TF_Barra*	Traffic flow at Barra
*TF_Costa*	Traffic flow at Costa

**Table 9 sensors-22-04485-t009:** CNN.

Beach	Input	Output	MAE/Mean	Std	Time/Mean	Std
Barra	1	1	6.79	0.41	56.42	16.38
	2	2	6.81	0.41	63.42	14.86
	3	3	7.28	0.58	55.81	26.43
	4	4	7.31	0.55	60.14	17.59
	5	5	8.05	0.59	63.27	18.97
	6	6	8.47	0.53	66.22	29.52
Costa	1	1	6.74	0.23	59.40	16.87
	2	2	6.88	0.47	66.62	22.93
	3	3	6.96	0.45	70.03	18.63
	4	4	7.53	0.53	55.54	19.20
	5	5	7.73	0.32	62.59	18.89
	6	6	8.79	0.36	54.19	12.88

**Table 10 sensors-22-04485-t010:** LSTM.

Beach	Input	Output	MAE/Mean	Std	Time/Mean	Std
Barra	1	1	14.91	1.41	920.80	108.14
	2	2	13.84	2.39	1038.09	125.18
	3	3	13.71	1.94	1190.37	142.13
	4	4	13.83	1.61	1370.07	180.46
	5	5	13.94	3.35	1661.48	267.81
	6	6	14.73	1.17	1872.14	264.06
Costa	1	1	13.49	1.88	991.62	101.98
	2	2	14.29	1.73	1080.23	132.31
	3	3	14.29	1.92	1188.94	203.84
	4	4	14.47	2.07	1399.33	143.40
	5	5	14.92	1.16	1744.69	168.46
	6	6	14.97	1.23	1641.91	268.67

**Table 11 sensors-22-04485-t011:** AR LSTM.

Beach	Input	Output	MAE/Mean	Std	Time/Mean	Std
Barra	1	1	16.97	4.67	902.99	48.63
	2	2	9.21	1.42	649.23	146.02
	3	3	8.52	0.57	811.02	121.65
	4	4	8.34	0.35	1006.01	96.47
	5	5	8.80	0.90	1320.48	218.51
	6	6	9.51	1.60	1413.75	370.97
Costa	1	1	17.89	6.14	898.62	65.68
	2	2	8.76	0.63	730.98	154.28
	3	3	8.74	1.20	891.32	129.97
	4	4	8.66	0.95	1014.53	105.89
	5	5	9.29	1.02	1254.10	265.74
	6	6	11.92	2.67	1386.19	298.42

## Data Availability

The dataset is available at https://figshare.com/s/d324f5be912e7f7a0d21 (accessed on 4 April 2022).

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
