# Peer review of "Road Traffic Forecast Based on Meteorological Information through Deep Learning Methods"

_sensors, 2022, doi:10.3390/s22124485_

Round 1

Reviewer 1 Report

This paper presents a work that evaluates different machine learning algorithms to forecast road traffic flows to two beaches areas in Portugal. The study uses as inputs: (1) traffic radar information and (2) meteorological data provided by the Portuguese Institute of Sea and Atmosphere (IPMA). The authors have created their own dataset that it is available at figshare portal. The algorithms tested are CNN, LSTM and AR-LSTM. The best results are obtained for 10 minutes intervals.

I have several concerns about this paper:

1    (1) The state of the art is a brief description of several research works related with traffic and road speed forecasting in recent years but there is not a critical analysis, neither conclusions to justify the work done by the authors in their own research. What are the main differences? What are the contributions of this new research work to the state of the art?

2  (2) The parameters selection in the machine learning algorithms is not justified, for example, in CNN the authors use a kernel size equals to 6, 64 filter maps, 200 epochs…

3  (3) The selection of training – validation – testing percentages are not justified.

4   (4) The best results for learning and forecasting are obtained for intervals of 10 minutes. Is this interval useful to be used in a real system in order to improve traffic management?

 Some typos:

Title: “mereological”

Line 82: In [?]

Line 304: Figure ??

Line 327: Figure ??

Author Response

This paper presents a work that evaluates different machine learning algorithms to forecast road traffic flows to two beaches areas in Portugal. The study uses as inputs: (1) traffic radar information and (2) meteorological data provided by the Portuguese Institute of Sea and Atmosphere (IPMA). The authors have created their own dataset that is available at figshare portal. The algorithms tested are CNN, LSTM and AR-LSTM. The best results are obtained for 10 minutes intervals.

First of all, we would like to express our sincere appreciation to reviewer 1 for his help in improving the paper.

I have several concerns about this paper:

1    (1) The state of the art is a brief description of several research works related with traffic and road speed forecasting in recent years but there is not a critical analysis, neither conclusions to justify the work done by the authors in their own research. What are the main differences? What are the contributions of this new research work to the state of the art?

Subsection text was edited and specific comments were included. A new column was added to table 2, indicating the limitations with respect to individual implementation. Finally a new paragraph was created summarizing the differences between the previous works and present contribution.

2  (2) The parameters selection in the machine learning algorithms is not justified, for example, in CNN the authors use a kernel size equals to 6, 64 filter maps, 200 epochs…

Each model was constructed considering the best configuration, defined as the result of an optimization phase by using the hyperparameters options (neurons, activation function, optimizer function, dropout, batch size, filter map, kernel size). Besides, each model was trained to find the best number of epochs, considering the limit of 200 epochs.

3  (3) The selection of training – validation – testing percentages are not justified.

The explanation of the reasons for dividing the dataset in that manner was included in section 3.4.

4   (4) The best results for learning and forecasting are obtained for intervals of 10 minutes. Is this interval useful to be used in a real system in order to improve traffic management?

This interval is useful to be used in real systems as it provides feedback on the current state of the traffic. For example, if a user chooses to take a short trip and, with access to the data for the next 10 minutes, he will be able to change the chosen route. By making users choose the freest paths, we are positively influencing traffic management. Other intervals could be used (for example, two hours), and these would be more suitable for longer trips.

 Some typos:

Title: “mereological”

It was fixed.

Line 82: In [?]

The reference was fixed.

Line 304: Figure ??

The reference was fixed.

Line 327: Figure ??

The reference was fixed.

Reviewer 2 Report

The manuscript requires revision based on identified issues present in the proposed work.

1. Firstly, is it mereological information or meteorological sensor information?

2. Add full abbreviation also LSTM, ARLSTM, and CNN in the Abstract.

3. Revise the abstract carefully and show what is your main work in this paper?

4. In Section 1, add motivation, contribution, and organization in a separate sub-section. Now version manuscript has not sufficient motivational work based on your title.

5. What is the role of section 2? You can merge this section into sections 1 or 3. Carefully check various abbreviations are available without full form. Provide full form.

6. Use the proper method for citations such as [last name et al. [1 or 2 or 3].

7. In Table 1, add the column as years and limitations. Then show how to mitigate these limitations with the proposed work.

8. Figure 1. Data Sources?

9. Figure 2. Methodology? Revise it as Methodology for……………. Improve the visibility of figure 2.

10. What is the functionality of LSTM and CNN in the proposed work? Mention it in section 4 as algorithms.

11. Line 304, Figure??

12. Add an abbreviation table before section 3 for a better understanding of the manuscript.

13.  Add contents in subsection 4.4.3.

14. The authors should remove the grammatical mistakes and typos in the paper.

15. Experiment evaluation details add in Section 5 for a better understanding of the manuscript work.

16. Add future scope in the conclusion.

17. Cross-reference all citations and ensure that they match accordingly. The reference paper format should be uniform. I have identified some of the references with missing details like page numbers, volume numbers, issue numbers, etc. Recent reference must be added, the following is recommended:

·          Deep learning for road traffic forecasting: Does it make a difference?

·          A deep learning-based IoT-oriented infrastructure for secure smart city.

·          Multistep traffic speed prediction: A deep learning based approach using latent space mapping considering spatio-temporal dependencies

Author Response

Author's Reply to the Review Report (Reviewer 2)

The manuscript requires revision based on identified issues present in the proposed work.

First of all, we would like to express our sincere appreciation to reviewer 2 for his help in improving the paper, especially for his suggestion in terms of references to enrich the related work. 

  1. Firstly, is it mereological information or meteorological sensor information?

The typo was fixed.

  1. Add full abbreviation also LSTM, ARLSTM, and CNN in the Abstract.

Abbreviations were added.

  1. Revise the abstract carefully and show what is your main work in this paper?

The abstract was edited and our main work was highlighted.

  1. In Section 1, add motivation, contribution, and organization in a separate sub-section. Now version manuscript has not sufficient motivational work based on your title.

The subsection division was included, and given the fact that the motivation was not clear, we deeply edited the section highlighting the motivation and adding two research questions.

  1. What is the role of section 2? You can merge this section into sections 1 or 3. Carefully check various abbreviations are available without full form. Provide full form.

We agree with reviewer 2 and section 2 and 3 were merged, and abbreviations were edited.

  1. Use the proper method for citations such as [last name et al. [1 or 2 or 3].

Citations were edited.

  1. In Table 1, add the column as years and limitations. Then show how to mitigate these limitations with the proposed work.

Column was created as suggested, and a paragraph was created at the end of the section to summarize the overall limitations, and to justify our own approach.

  1. Figure 1. Data Sources?

The label was changed.

  1. Figure 2. Methodology? Revise it as Methodology for……………. Improve the visibility of figure 2.

Both the figure and the label were edited.

  1. What is the functionality of LSTM and CNN in the proposed work? Mention it in section 4 as algorithms.

Current work included the implementation and the performance assessment  of some CNN, LSTM and AR LSTM algorithms to forecast highway traffic. The performance analysis allowed us to verify that the CNN-based algorithm has a better performance compared to the other technologies algorithms (LSTM and AR LSTM), and that the best strategy is to use individual periods (10 minutes), either for learning or for prediction. The lessons learned from the present work are currently being used in the development of a real-time TSE that will feed the PASMO project dashboard.

 Some text here.

  1. Line 304, Figure??

Reference was edited.

  1. Add an abbreviation table before section 3 for a better understanding of the manuscript.

The acronym table was added.

  1. Add contents in subsection 4.4.3.

Our approach was to condense the technological descriptions so that the subsection that merged section and section 3 would not be too long.

  1. The authors should remove the grammatical mistakes and typos in the paper.

Text was deeply changed, and the typos were removed.

  1. Experiment evaluation details add in Section 5 for a better understanding of the manuscript work.

Section 5  text ( now section 4) was edited and further details were added about the text conditions.

  1. Add future scope in the conclusion.

Future work was detailed in the conclusion section.

  1. Cross-reference all citations and ensure that they match accordingly. The reference paper format should be uniform. I have identified some of the references with missing details like page numbers, volume numbers, issue numbers, etc. Recent reference must be added, the following is recommended:
  • Deep learning for road traffic forecasting: Does it make a difference?
  • A deep learning-based IoT-oriented infrastructure for secure smart city.
  • Multistep traffic speed prediction: A deep learning based approach using latent space mapping considering spatio-temporal dependencies

We appreciate the suggestions of papers that we had not previously found and that we ended up referencing partially. We reviewed the references bib content.

Round 2

Reviewer 1 Report

The authors have addressed my major concerns.